# Reconstitution reveals two paths of force transmission through the kinetochore

**Grace E Hamilton[1], Luke A Helgeson[1], Cameron L Noland[2], Charles L Asbury[3]\*, Yoana N Dimitrova[2]\*, Trisha N Davis[1]\***

[1]Department of Biochemistry, University of Washington, Seattle, United States; [2]Department of Structural Biology, Genentech Inc, South San Francisco, United States; [3]Department of Physiology and Biophysics, University of Washington, Seattle, United States

**Abstract** Partitioning duplicated chromosomes equally between daughter cells is a microtubule-mediated process essential to eukaryotic life. A multi-protein machine, the kinetochore, drives chromosome segregation by coupling the chromosomes to dynamic microtubule tips, even as the tips grow and shrink through the gain and loss of subunits. The kinetochore must harness, transmit, and sense mitotic forces, as a lack of tension signals incorrect chromosome-microtubule attachment and precipitates error correction mechanisms. But though the field has arrived at a 'parts list' of dozens of kinetochore proteins organized into subcomplexes, the path of force transmission through these components has remained unclear. Here we report reconstitution of functional *Saccharomyces cerevisiae* kinetochore assemblies from recombinantly expressed proteins. The reconstituted kinetochores are capable of self-assembling in vitro, coupling centromeric nucleosomes to dynamic microtubules, and withstanding mitotically relevant forces. They reveal two distinct pathways of force transmission and Ndc80c recruitment.

**\*For correspondence:**
casbury@uw.edu (CLA);
dimitry4@gene.com (YND);
tdavis@uw.edu (TND)

## Introduction

Chromosome segregation is a complex imperative faced by all eukaryotes, as failure to accurately distribute genetic material to daughter cells in mitosis and meiosis causes potentially lethal aneuploidy (*Cimini et al., 2001*; *Cimini, 2008*; *Santaguida and Amon, 2015*). Eukaryotic cells rely on a network of protein complexes (*Figure 1A* and *Table 1*) to tether centromeres to the dynamic spindle microtubules that pull them to opposite poles of the dividing cell (*Musacchio and Desai, 2017*).

This network, the kinetochore, mediates the vital process of chromosome segregation by coupling microtubule dynamics to chromosome movement. The inner kinetochore binds the centromere; the outer kinetochore binds microtubules. A massive molecular machine, the kinetochore must harness the force of depolymerizing microtubules, transmit this force to the centromere, and sense aberrant attachments that fail to generate force. The kinetochore also serves as a regulatory hub for detection of unattached kinetochores and destabilization of improper kinetochore-microtubule attachments (*Musacchio, 2011*). Tension is the signal of correct attachment (i.e., biorientation) (*Nicklas and Ward, 1994*), and it directly stabilizes kinetochore-microtubule connections (*Akiyoshi et al., 2010*). But which of the myriad kinetochore proteins actually take part in force transmission and how much tension the kinetochore is under are unresolved questions.

Estimates of mitotic forces experienced by kinetochores in vivo span orders of magnitude (*Nicklas, 1988*; *Fisher et al., 2009*; *Ye et al., 2016*). In the case of budding yeast, each kinetochore contacts a single microtubule and is estimated to experience forces between 0.2 and 16 pN during metaphase (*Fisher et al., 2009*; *Chacón et al., 2014*; *Suzuki et al., 2016*). Purified kinetochore particles isolated from budding yeast couple to microtubule plus ends with an average rupture strength in this range: 9.1 pN (*Akiyoshi et al., 2010*; *Miller et al., 2016*). Previous studies using purified

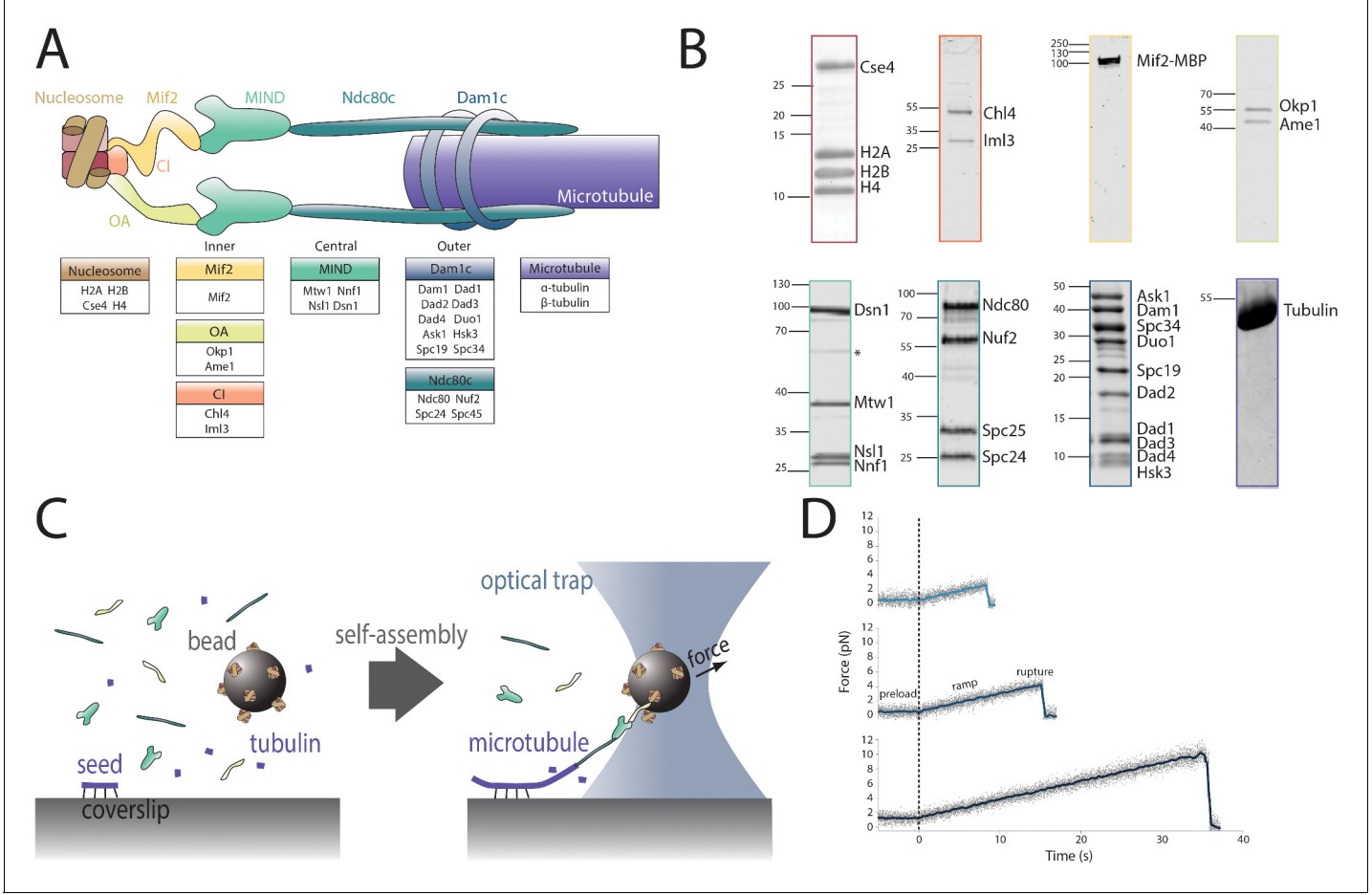

**Figure 1.** Reconstitution of a kinetochore from individually purified parts and an optical trap-based assay to test for self-assembly of functional chains of kinetochore subcomplexes. (A) Schematic of the protein complexes of the budding yeast kinetochore. (B) Coomassie-stained SDS-PAGE gel of heterologously expressed budding yeast kinetochore proteins. The asterisk indicates a contaminating *E. coli* protein or degradation product. (C) Schematic of the optical trap assay used to test for assembly and microtubule attachment prior to quantification of load-bearing ability. (D) Representative force vs. time traces for ruptures in the force-ramp assay.

The online version of this article includes the following source data and figure supplement(s) for figure 1:

**Source data 1.** Plasmids used in this study.

**Figure supplement 1.** Schematic diagram, drawn approximately to scale, showing two possible bead-microtubule configurations.

proteins have concluded that (1) Ndc80c and Dam1c are both capable of coupling to microtubules independently, although Dam1c improves microtubule coupling by Ndc80c; (2) this process is regulated by Aurora B kinase (Ipl1 in budding yeast) (*Cheeseman et al., 2002*; *Cheeseman et al., 2006*; *Caldas et al., 2013*; *Sarangapani et al., 2013*; *Umbreit et al., 2014*; *Kim et al., 2017*); (3) Ndc80c transmits force to MIND (*Kudalkar et al., 2015*; *Dimitrova et al., 2016*), and (4) the kinetochore as a whole behaves as a tension-sensitive catch bond (*Akiyoshi et al., 2010*). Load-bearing tip-couplers can form in vitro by self-assembly of recombinant Ndc80c (*Powers et al., 2009*), MIND (*Kudalkar et al., 2015*), and Dam1c (*Asbury et al., 2006*), demonstrating that these components are sufficient to recapitulate a fundamental function of the outer kinetochore. Relatively little is known about how force is transmitted to the centromeric DNA. It is not even certain which inner kinetochore proteins are involved.

There are only two essential inner kinetochore protein complexes in budding yeast: Mif2 and OA (Okp1/Ame1) (*Meeks-Wagner et al., 1986*; *Ortiz et al., 1999*; *Pot et al., 2005*; *Hara and Fukagawa, 2017*; *Table 1*). Mif2, the budding yeast homolog of human CENP-C, is reported to interact with centromeric nucleosomes (*Xiao et al., 2017*), OA (*Hornung et al., 2014*), and MIND (*Hornung et al., 2014*; *Dimitrova et al., 2016*). Homologous to human CENP-QU, *S. cerevisiae* OA

**Table 1.** Proteins of the kinetochore.

| *S. cerevisiae* | *H. sapiens* |
|---|---|
| Dam1c/DASH | |
| Ask1 (Associated with spindles and kinetochores) | |
| Dad1 (Duo1 and Dam1 interacting) | |
| Dad2 (Duo1 and Dam1 interacting) | |
| Dad3 (Duo1 and Dam1 interacting) | Higher eukaryotic analog is the Ska complex |
| Dad4 (Duo1 and Dam1 interacting) | |
| Dam1 (Duo1 and Mps1 interacting) | |
| Duo1 (Death upon overproduction) | |
| Hsk3 (Helper of Ask1) | |
| Spc19 (Spindle pole component) | |
| Spc34 (Spindle pole component) | |
| | **Ska Complex** |
| | Ska1 |
| Functional analog is Dam1c | Ska2 |
| | Ska3 |
| **Spc105c** | **KNL1 complex** |
| Spc105 (Spindle component) | KNL1 |
| Kre28 (Killer toxin resistant) | Zwint-1 |
| **Ndc80c** | **Ndc80c** |
| Ndc80 (Nuclear division cycle) | Hec1 |
| Nuf2 (Nuclear filamentous protein) | Nuf2 |
| Spc24 (Spindle pole component) | Spc24 |
| Spc25 (Spindle pole component) | Spc25 |
| **MIND** | **Mis12c** |
| Mtw1 (Mis Twelve-like) | Mis12 |
| Dsn1 (Dosage suppressor of *NNF1*) | Dsn1 |
| Nnf1 (Necessary for nuclear function) | Pmf1 |
| Nsl1 (*NNF1* synthetic lethal) | Nsl1 |
| **Cnn1c** | **CENP-TWSX** |
| Cnn1 (Co-purified with Nnf1) | CENP-T |
| Wip1 (W-like protein) | CENP-W |
| Mhf1 (Mph1-associated histone-fold protein) | CENP-S |
| Mhf2 (Mph1-associated histone-fold protein) | CENP-X |
| **OA** | **CENP-QU** |
| Okp1 (Outer kinetochore protein) | CENP-Q |
| Ame1 (Associated with microtubules and essential) | CENP-U |
| **Mif2** | **CENP-C** |
| Mif2 (Mitotic fidelity of chromosome transmission) | CENP-C |
| **CI** | **CENP-NL** |
| Chl4 (Chromosome loss) | CENP-N |
| Iml3 (Increased minichromosome loss) | CENP-L |

*Table 1 continued on next page*

*Table 1 continued*

| S. cerevisiae | H. sapiens |
| --- | --- |
| NN | |
| Nkp1 (Non-essential kinetochore protein) | No human homolog |
| Nkp2 (Non-essential kinetochore protein) | |
| CM | CENP-OP |
| Ctf19 (Chromosome transmission fidelity) | CENP-O |
| Mcm21 (Mini-chromosome maintenance) | CENP-P |
| No fungal homolog | CENP-R |
| Ctf3c | CENP-HIKM |
| Ctf3 (Chromosome transmission fidelity) | CENP-H |
| Mcm16 (Mini-chromosome maintenance) | CENP-I |
| Mcm22 (Mini-chromosome maintenance) | CENP-K |
| No fungal homolog | CENP-M |
| Centromeric histone | Centromeric histone |
| Cse4 (Chromosome segregation) | CENP-A |

interacts with the centromeric histone Cse4 and MIND, the scaffold that bridges inner and outer kinetochore (*Hornung et al., 2014*; *Dimitrova et al., 2016*; *Anedchenko et al., 2019*; *Fischböck-Halwachs et al., 2019*). Because force transmission is a vital function of the kinetochore, we hypothesized that either Mif2 or OA must transmit force through the inner kinetochore.

Here, using recombinantly purified *S. cerevisiae* proteins, we reconstitute kinetochore assemblies capable of tethering centromeric nucleosomes to dynamic microtubule plus-ends. We show that the assemblies are load-bearing, in the sense that they sustain tensile forces in the mitotically relevant piconewton range. In doing so, we demonstrate that there are at least two possible paths of force transmission through the inner kinetochore: the Mif2 and OA protein complexes. Each of these components can independently form load-bearing interactions with both MIND and the centromeric nucleosome.

## Results

### Assay to test quantitatively for spontaneous self-assembly of purified kinetochore subcomplexes

The proteins of the *S. cerevisiae* kinetochore are organized into subcomplexes (*Figure 1A*). We have recombinantly expressed and purified seven of these subcomplexes in *E. coli* and isolated bovine brain tubulin to allow reconstitution and mechanical testing of functional kinetochore assemblies in vitro (*Figure 1B*). Previously, we showed that the microtubule-binding Ndc80c and Dam1c complexes, when introduced free in solution, will associate with bead-bound MIND to form a load-bearing attachment to a dynamic microtubule tip (*Kudalkar et al., 2015*). We have now extended this approach to test longer protein chains, including both inner and outer kinetochore components. In order to test a protein chain containing multiple components, one subcomplex carrying a His$_6$-tag was bound directly to polystyrene microbeads via anti-His antibodies (*Figure 1C*). The remaining subcomplexes, which did not have His$_6$-tags, were added free in solution and bound to the beads only indirectly, by assembling together with the directly tethered, His$_6$-tagged subcomplex. Using a laser trap to manipulate individual beads, the kinetochore assemblies were tested for their ability to

**Table 2.** Plasmids used in this study.

| Protein complex | Plasmid name | Names used in this paper | Proteins expressed* | Vector | References |
|---|---|---|---|---|---|
| Mif2 | Sc_Mf_7 | Mif2 | Mif2-linker-(27-392)MBP-6XHis** | pLIC | This study |
| | pGH52 | Mif2 | Mif2-linker-(27-392)MBP** | pLIC | This study |
| | Sc_Mf_5B | ΔN-Mif2 | (41-549)Mif2-linker-(27-392)MBP | pLIC | This study |
| OA | pGH3 | OA | Ame1-6XHis, Okp1 | pST39 | This study |
| | pGH4 | OA | Ame1-FLAG, Okp1 | pST39 | This study |
| | pGH42 | ΔN-OA | (21-324)Ame1-FLAG, Okp1 | pST39 | This study |
| | pGH15 | ΔN-OA | (21-324)Ame1-6XHis, Okp1 | pST39 | This study |
| MIND | pGH63 | 2D-MIND | 6XHis-linker-Nsl1, S240D, S250D-Dsn1, Mtw1, Nnf1 | pST39 | This study |
| | pGH62 | 2D-MIND | FLAG-Nsl1, S240D, S250D-Dsn1, Mtw1, Nnf1 | pST39 | This study |
| | pGH46 | MIND | Nsl1, FLAG-Dsn1, Mtw1, Nnf1 | pST39 | This study |
| Ndc80c | pJT048 | Part of Ndc80c | Spc24-Flag, Spc25 | pRSFDuet | *Kudalkar et al., 2015* |
| | pEM033 | Part of Ndc80c | Spc24-6XHis, Spc25 | pRSFDuet | *Scarborough et al., 2019* |
| | Ndc80/Nuf2 | Part of Ndc80c | Nuf2, Ndc80 | pETDuet | *Wei et al., 2005* |
| Dam1c | pJT044 | Dam1c | Dad1, Duo1, Spc34-FLAG, Dam1, Hsk3 and Dad4, Dad3, Dad2, Spc19, Ask1‡ | pST39 | *Umbreit et al., 2014* |
| CI | pGH58 | CI | FLAG-Chl4, Iml3 | pLIC | This study |
| Histones | pScKl2 | Cse4-NCP | *K.lactis* 6XHis-H2A, *K. lactis* 6XHis-H2B, Cse4, *K. lactis* 6XHis-H4 | pLIC | *Migl et al., 2020* |
| | pScKl4 | H3-NCP | H3, 6XHis-H2A, H2B. *K.lactis* 6XHis-H4 | pLIC | *Migl et al., 2020* |
| | pScHT4 | Cse4(1-50) | 6XHis-MBP-(1-50)Cse4 | pLIC | This study |

couple to an individual, assembling microtubule tip. If they attached, then their coupling strength was subsequently measured by increasing the force until the bead detached from the tip (*Asbury et al., 2006*; *Powers et al., 2009*; *Akiyoshi et al., 2010*; *Franck et al., 2010*; *Figure 1D*). Protein chains that withstood the 1-pN preload force were considered load-bearing, and distributions of rupture force were used to compare relative mechanical strength. Because bond strength is determined by both kinetic and thermodynamic properties and varies depending on the loading rate (*Evans and Ritchie, 1997*; *Merkel et al., 1999*; *Bustamante et al., 2004*), loading rates and protein concentrations were held constant in all experimental conditions to allow meaningful comparison between the mechanical strengths of different protein linkages. In negative controls, microtubule coupling was abrogated by omitting one of the subcomplexes. Because these experiments were not conducted under single-molecule conditions, many kinetochore protein assemblies on each bead were presumably contacting each individual microtubule tip (*Figure 1—figure supplement 1*); this arrangement mimics the physiological situation in budding yeast, where multiple microtubule-binding elements of the outer kinetochore form a multivalent attachment to a single microtubule.

## OA forms microtubule attachments through MIND and Ndc80c

Both of the two essential inner kinetochore protein complexes, Mif2 and OA, bind directly to MIND (*Hornung et al., 2014*; *Dimitrova et al., 2016*). Thus we began by asking whether OA could form a microtubule attachment through MIND and Ndc80c. With OA bound to the polystyrene beads, and MIND and Ndc80c added in solution (at 10 nM), the vast majority of beads coupled to assembling microtubule tips, indicating that functional, OA-based co-complexes spontaneously assembled in vitro (*Figure 2A*). No microtubule attachments were observed if MIND was omitted from the assay, demonstrating that OA must interact specifically through MIND – and not with Ndc80c or the microtubule directly – in order for the microtubule-coupling assemblies to form. Likewise, very few beads

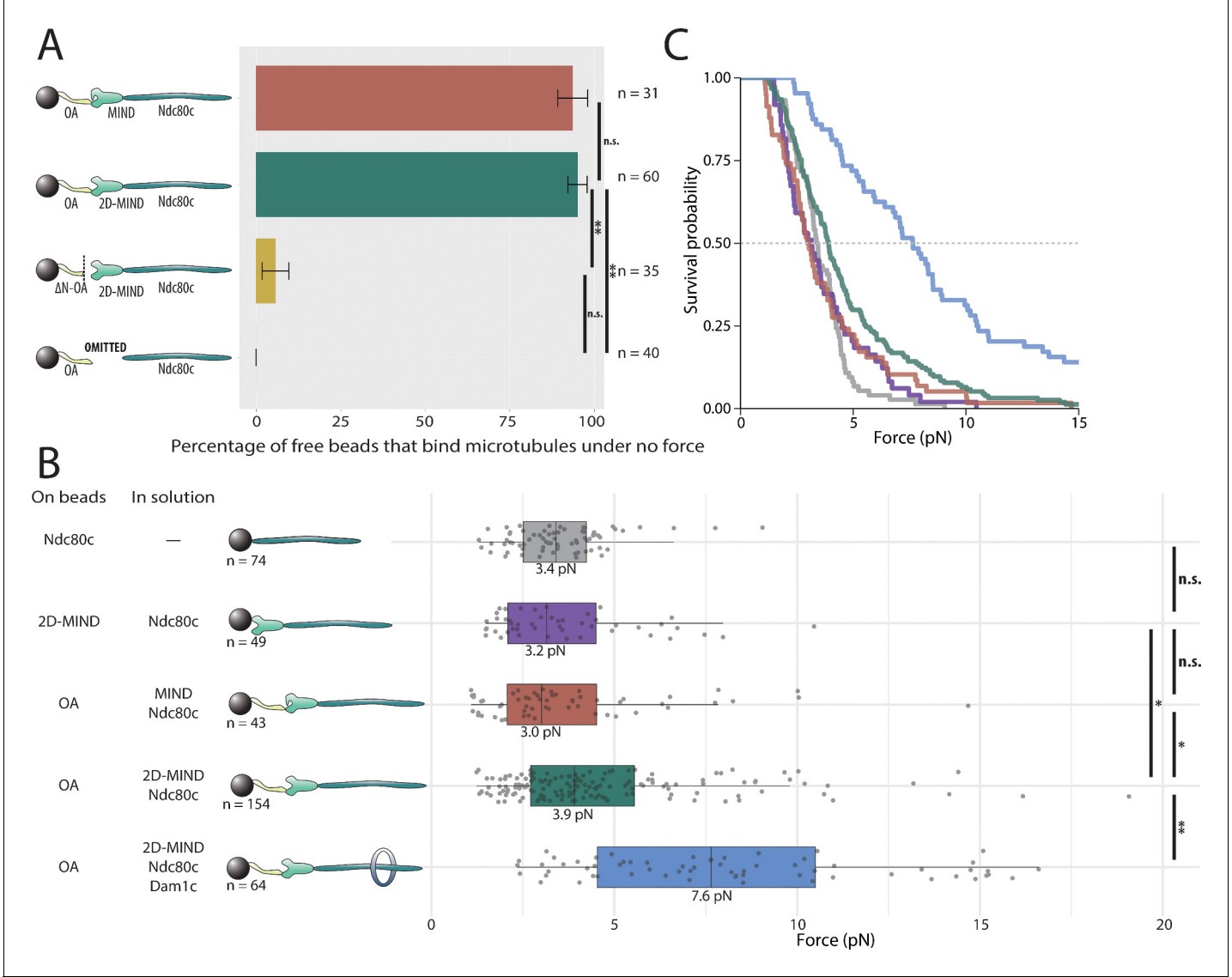

**Figure 2.** OA-based chains assemble spontaneously and form load-bearing attachments to dynamic microtubules. (**A**) Percentages of free beads that bound microtubules under no external force. Error bars indicate the standard error of the sample proportion. Barnard's test was used to compare contingency tables. n.s. indicates p>0.05. * indicates p<0.05. ** indicates p<0.01. (**B**) Boxplot of rupture forces observed with reconstituted kinetochores of increasing length. Each shaded circle is an individual rupture event. Boxes extend from the lower quartile to the upper quartile. Whiskers extend to 1.5 times the interquartile range beyond each quartile. A Kolmogorov-Smirnov test was used to compare probability distributions and calculate p-values. n.s. indicates p>0.05. * indicates p<0.05. ** indicates p<0.01. (**C**) Survival curves for Ndc80c linkages (grey), MIND/Ndc80c linkages (purple), OA/MIND/Ndc80c linkages (pink), OA/2D-MIND/Ndc80c linkages (turquoise), and OA/2D-MIND/Ndc80c/Dam1c linkages (blue). The dashed horizontal line indicates 50% survival (median rupture force). Raw data of all rupture events are included in *Figure 2—source data 1*. Exact p-values are included in *Figure 2—source data 2*.

The online version of this article includes the following source data for figure 2:

**Source data 1.** Raw rupture force data for OA-based linkers.

**Source data 2.** Exact p-values for all comparisons of rupture force distributions.

attached (6%) when they were decorated with a truncated mutant OA complex (ΔN-OA), lacking the N-terminal 20 residues of Ame1 that bind the MIND complex (*Hornung et al., 2014*; *Dimitrova et al., 2016*; *Figure 2A*). Together with the prior observation that MIND itself has no microtubule-binding activity (*Kudalkar et al., 2015*), these results indicate that a three-component chain, OA/MIND/Ndc80c, spontaneously assembles to couple OA <u>in</u>directly to the microtubule tip.

This arrangement is consistent with localization dependencies seen in vivo (*De Wulf et al., 2003*), co-immunoprecipitation (co-IP) dependencies in budding yeast (*De Wulf et al., 2003*; *Lang et al., 2018*), and with previous in vitro reconstitutions of metazoan kinetochore proteins (*Cheeseman et al., 2006*; *Weir et al., 2016*). (Although it should be noted that human homologs of OA are reported to directly bind microtubules and not the Mis12c/MIND *Pesenti et al., 2018*, suggesting interspecific differences in how this conserved subcomplex is utilized.)

Phosphorylation of two sites on Dsn1 has been shown to partially alleviate an intra-complex autoinhibition, thereby allowing MIND to interact more stably with both OA and Mif2 (*Dimitrova et al., 2016*; *Petrovic et al., 2016*). However, we found that both the wild-type MIND complex and a phosphomimetic mutant, 2D-MIND, carrying aspartic acid substitutions at the two target phosphorylation sites on Dsn1 (S240D, S250D), supported the attachment of a similarly high fraction of OA-coated beads to assembling microtubule tips (*Figure 2A*). This observation suggests that the intra-complex autoinhibition within MIND does not completely abrogate its interaction with OA.

## OA forms a load-bearing attachment to MIND

Because tip-coupled kinetochores must support piconewton-scale loads in vivo, we asked whether the reconstituted OA/MIND/Ndc80c chains could support such loads in vitro. In order to quantify their load-bearing ability, we used a force-ramp assay (*Franck et al., 2010*). After attachment of a bead to a single growing microtubule tip and application of a weak preload force (~1 pN), the force was gradually increased (at 0.25 pN s$^{-1}$) until the attachment ruptured. Rupture is a stochastic event, and therefore rupture forces measured for populations of identical protein linkages always exhibit significant variability (*Evans and Ritchie, 1997*; *Merkel et al., 1999*; *Akiyoshi et al., 2010*; *Kudalkar et al., 2015*; *Helgeson et al., 2018*). For this reason, dozens of ruptures were collected for every condition tested. The median rupture strength of tip-attachments formed by OA/MIND/Ndc80c chains was 3.0 pN (*Figure 2B,C*), indicating that both the OA/MIND and MIND/Ndc80c interfaces can bear load. When the phosphomimetic 2D-MIND was used instead of wild-type MIND, the median rupture strength of the OA/2D-MIND/Ndc80c chains increased slightly, to 3.9 pN. This observation suggests that the autoinhibition within the MIND complex slightly weakens the load-bearing capacity of the OA/MIND interface.

While rupture strengths by themselves do not provide information about which protein-protein interface breaks during each rupture, some inferences can be made from comparative analysis. For example, previously published work demonstrates that MIND/Ndc80c-mediated microtubule attachments do not differ in strength from Ndc80c-mediated attachments (*Kudalkar et al., 2015*), implying that the MIND/Ndc80c interface is mechanically stronger than the Ndc80c/microtubule interface. This conclusion was additionally supported by the prior observation that adding Dam1c, which specifically strengthens the Ndc80c/microtubule interface, significantly increases the strength of MIND/Ndc80c-mediated attachments (*Kudalkar et al., 2015*). In order to determine if the OA/MIND interface is likewise mechanically strong relative to the Ndc80c/microtubule interface, we specifically strengthened the latter by adding Dam1c in solution to our rupture force assay. OA/MIND/Ndc80c-mediated attachments nearly doubled in strength, to a median rupture force of 7.6 pN, with the addition of Dam1c (*Figure 2B,C*). This observation suggests that the OA/MIND interface, like the MIND/Ndc80c interface, is mechanically strong relative to the Ndc80c/microtubule interface.

## Mif2 forms microtubule attachments through MIND and Ndc80c

We then asked if Mif2 could form microtubule attachments through MIND and Ndc80c. Mif2 was bound to polystyrene beads, while wild-type MIND and Ndc80c were added in solution. A substantial fraction of the beads, 45%, coupled to microtubule tips, demonstrating that functional, three-component Mif2/MIND/Ndc80c chains spontaneously assembled in vitro (*Figure 3A*). Hypothesizing that intra-complex autoinhibition within MIND might be responsible for the 55% of beads that failed to bind microtubules, we repeated the same experiment, but using the phosphomimetic mutant, 2D-MIND, instead of wild-type MIND. With this substitution, the fraction of Mif2-coated beads capable of coupling to microtubules increased to 81% (*Figure 3A*), suggesting that intra-complex autoinhibition within MIND tightly regulates its interaction with Mif2. To confirm that MIND was essential for connecting Mif2 to Ndc80c, we omitted MIND from the assay. Beads coated in Mif2 did not couple to microtubules in the absence of MIND (*Figure 3A*), indicating that Mif2, like OA (*Figure 2A*),

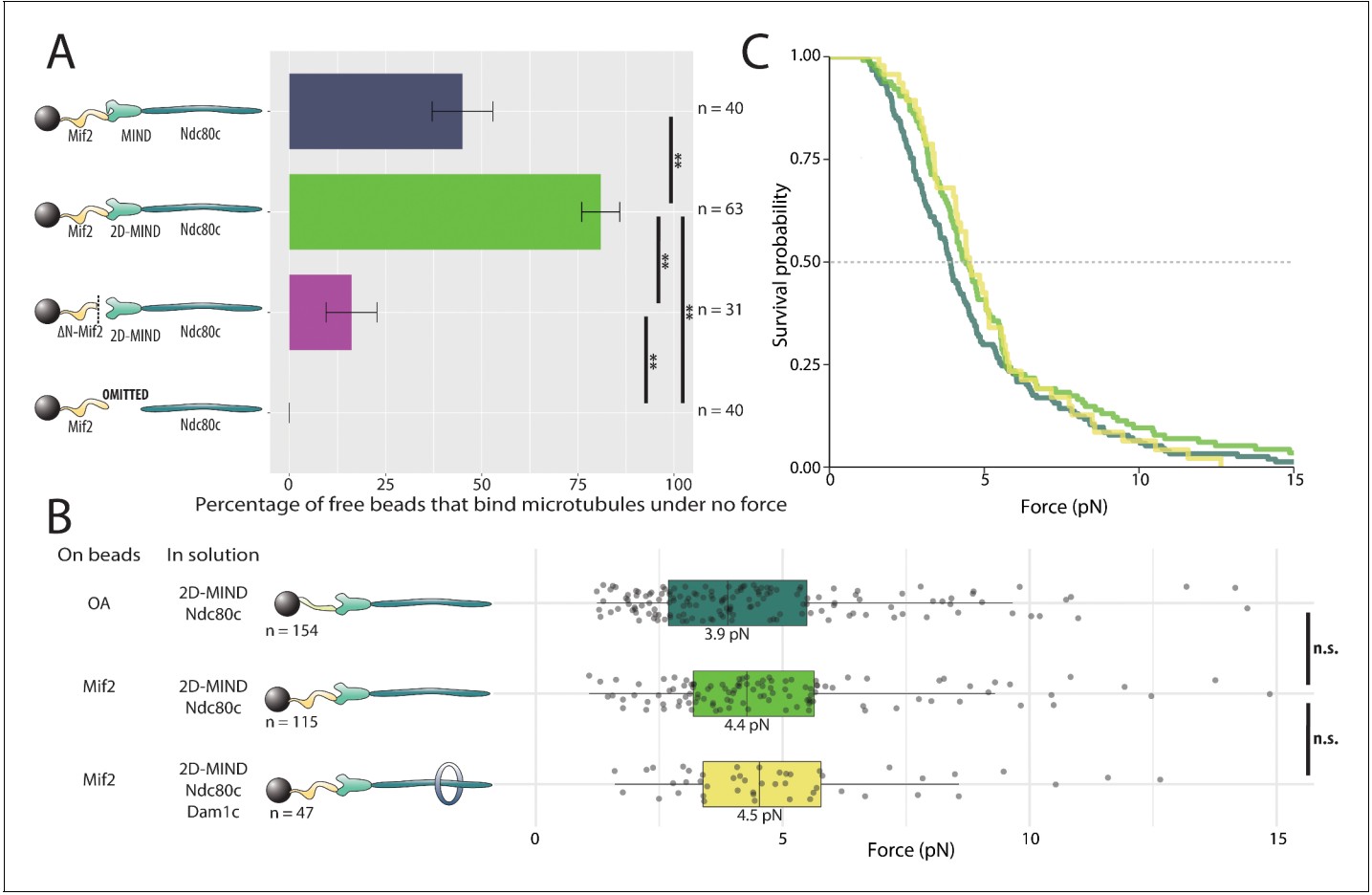

**Figure 3.** Mif2-based chains also assemble spontaneously and form load-bearing attachments to dynamic microtubule tips. (**A**) Percentages of free beads that bound microtubules under no force. Error bars indicate the standard error of the sample proportion. Barnard's test was used to compare contingency tables. n.s. indicates p > 0.05. * indicates p < 0.05. ** indicates p < 0.01. "2D" indicates that two phosphomimetic mutations (S240D, S250D) have been made to the MIND component Dsn1. (**B**) Boxplot of rupture forces observed with reconstituted kinetochores. Each shaded circle is an individual rupture event. Boxes extend from the lower quartile to the upper quartile. Whiskers extend to 1.5 times the interquartile range beyond each quartile. A Kolmogorov-Smirnov test was used to compare probability distributions and calculate p-values. n.s. indicates p > 0.05. * indicates p < 0.05. ** indicates p < 0.01. (**C**) Survival curves for OA/2D-MIND/Ndc80c linkages (turquoise) (repeated from *Figure 2* for comparison), Mif2/2D-MIND/Ndc80c linkages (green), and Mif2/2D-MIND/Ndc80c/Dam1c linkages (yellow). The dashed horizontal line indicates 50% survival (median rupture force). Raw data of all rupture events are included in *Figure 3—source data 1*".

The online version of this article includes the following source data for figure 3:

**Source data 1.** Raw rupture force data for Mif2-based linkers.

must bind directly to MIND, and not to Ndc80c or to microtubules. Unlike OA, however, Mif2 requires alleviation of autoinhibition within the MIND complex in order to bind tightly. For this reason, 2D-MIND was used in all subsequent experiments.

The conserved N-terminus of Mif2 is sufficient for binding to the Mtw1/Nnf1 head of the MIND complex (*Hornung et al., 2014*; *Dimitrova et al., 2016*). In order to probe the specific Mif2 domains involved in assembling functional microtubule-coupling chains, we purified a truncated version lacking the first 40 N-terminal residues, ΔN-Mif2. Only 16% of beads coated with ΔN-Mif2 interacted with microtubules (*Figure 3A*), suggesting that the N-terminal domain of Mif2, like the N-terminus of Ame1, is an important point of attachment between the inner kinetochore and MIND. But because a significant fraction of beads was still able to bind microtubules in the absence of the first 40 residues of Mif2, it seems likely that other amino acids also contribute to the Mif2/MIND interface. A region of looser conservation extends beyond the well-conserved first 40 residues at the N-terminus of Mif2 and its homologs (*Screpanti et al., 2011*).

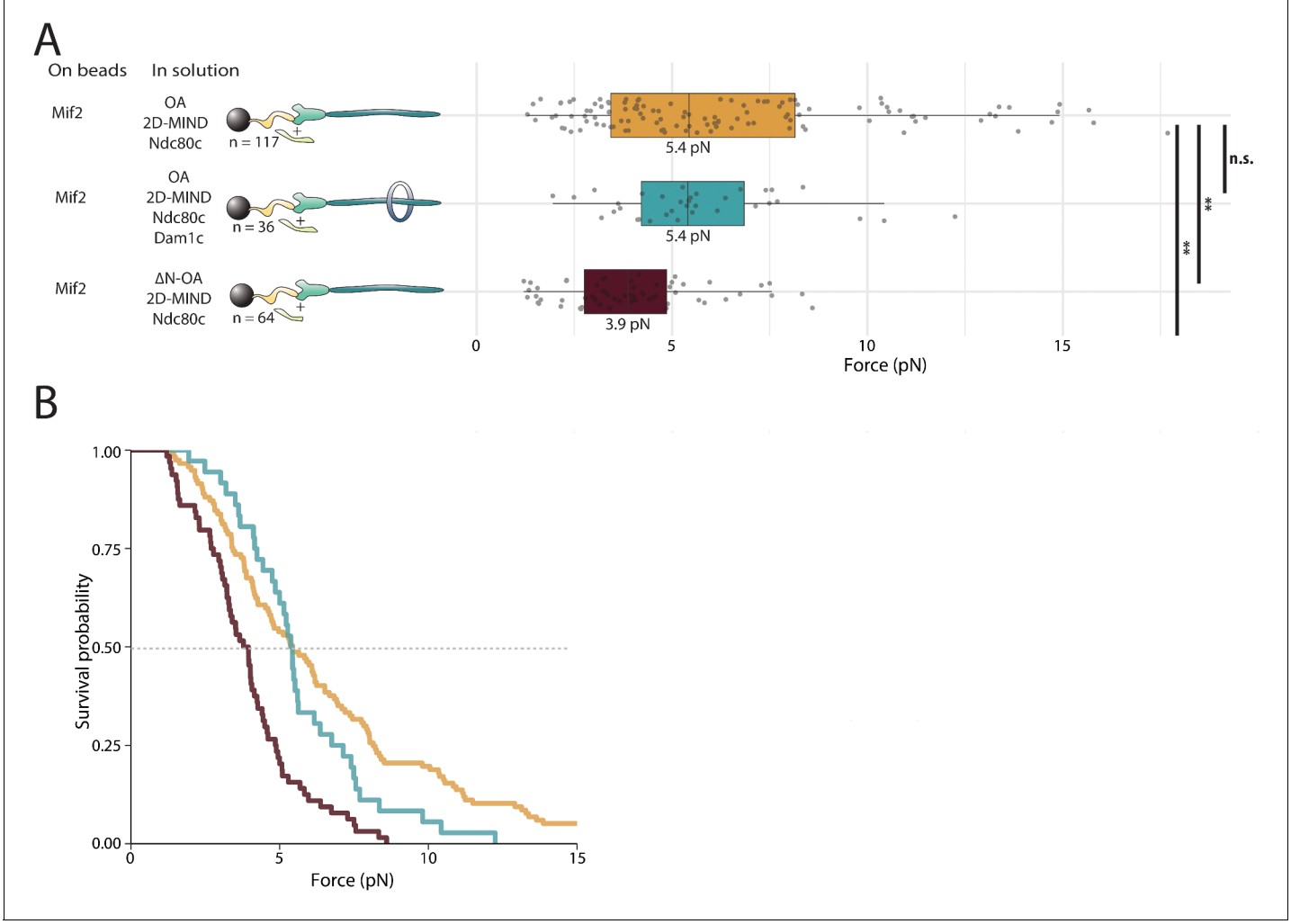

**Figure 4.** OA strengthens the Mif2/2D-MIND interface. (**A**) Boxplot of rupture forces observed with reconstituted kinetochores. Each shaded circle is an individual rupture event. Boxes extend from the lower quartile to the upper quartile. Whiskers extend to 1.5 times the interquartile range beyond each quartile. A Kolmogorov-Smirnov test was used to compare probability distributions and calculate p-values. n.s. indicates p>0.05. * indicates p<0.05. ** indicates p<0.01. (**B**) Survival curves for Mif2/OA/2D-MIND/Ndc80c linkages (orange), Mif2/OA/2D-MIND/Ndc80c/Dam1c linkages (teal), and Mif2/ΔN-OA/2D-MIND/Ndc80c (maroon). The dashed horizontal line indicates 50% survival (median rupture force). Raw data of all rupture events are included in *Figure 4—source data 1*.

The online version of this article includes the following source data and figure supplement(s) for figure 4:

**Source data 1.** Raw rupture force data for linkers containing both OA and Mif2.

**Figure supplement 1.** Mif2 does not strengthen the OA/2D-MIND interface.

## Mif2 forms a load-bearing attachment to MIND

To quantify the rupture strength of Mif2-based assemblies, we applied the force-ramp assay. Attachments mediated by the Mif2/2D-MIND/Ndc80c chains supported piconewton-scale loads, with a median rupture strength of 4.4 pN. Notably, their strength was not significantly different from that of our OA-based assemblies (*Figure 3B,C*). As explained above, the addition of Dam1c significantly strengthened the OA-based linkers, indicating that the OA/MIND interface is mechanically strong relative to the Ndc80c/microtubule interface. To ask if the same is true of the Mif2/MIND interface, we tested whether Mif2/2D-MIND/Ndc80c chains could be strengthened by the addition of Dam1c. In contrast to our results with OA, adding Dam1c had no significant effect (p=0.67) on the strength of the Mif2-based chains (*Figure 3B,C*). These observations suggest that the OA/MIND interface is

mechanically strong relative to the Ndc80c/microtubule attachment, but that the Mif2/MIND interface is not.

## OA strengthens the Mif2/2D-MIND interface

Having demonstrated that both Mif2 and OA can independently transmit force to MIND, we then asked if these two subcomplexes are stronger in combination. We added free OA to the Mif2/2D-MIND/Ndc80c chain, thereby increasing its median rupture force from 4.4 pN to 5.4 pN (*Figure 4A, B*). This increase suggests that OA is able to strengthen the Mif2/MIND interface, which we believe is relatively weak (based on our experiments with Dam1c). We then tested whether the Mif2-based chains with the addition of free OA could be further strengthened by adding free Dam1c. Adding Dam1c in solution did not change the 5.4 pN rupture force (*Figure 4A,B*), strongly suggesting that the Mif2/2D-MIND interface, although strengthened by the addition of OA, remains the primary site of rupture in Mif2/OA/2D-MIND/Ndc80c linkers.

In order to better understand *how* OA strengthens Mif2/2D-MIND/Ndc80c linkers, we replaced the free OA added in solutions with ΔN-OA, which is incapable of binding MIND (*Figure 2A*). In contrast to the increase in rupture force that we observed when free full-length OA was added to Mif2-based linkers, free ΔN-OA did not increase the strength of these linkers (*Figure 4A,B*). This observation shows that the MIND-binding activity of OA is essential for strengthening the Mif2/2D-MIND interface. There are two ways in which this could occur. Because '2D' phosphomimetic mutations on Dsn1 do not entirely alleviate autoinhibition within the MIND complex (*Dimitrova et al., 2016*), it is possible that OA strengthens the Mif2/2D-MIND interface by further alleviating that autoinhibition. Alternatively, if the free OA were able to bind both 2D-MIND and Mif2, then it could theoretically recruit additional MIND/Ndc80c to the linker, thereby increasing the valency and strength of the Ndc80c-microtubule interaction. Either scenario is consistent with ΔN-OA being unable to strengthen Mif2/2D-MIND/Ndc80c linkers. We favor the former theory, because we have no evidence of a direct interaction between OA and Mif2 in the absence of post-translational modifications, and because experiments with Dam1c (*Figure 4A,B*) suggest that the Mif2/2D-MIND interface is mechanically weak relative to the Ndc80c/microtubule interface.

In OA-based linkers, on the other hand, the OA/2D-MIND interface was not the weakest link, leading to the prediction that strengthening that interface should not increase the overall strength of OA-based chains. Indeed, the addition of Mif2 in solution did not strengthen OA/2D-MIND/Ndc80c-mediated attachments (*Figure 4—figure supplement 1*), consistent with our earlier inference that the Ndc80c/microtubule interface is usually the site of rupture in our OA-based assemblies.

## Both OA and Mif2 assemble with centromeric nucleosomes

Previous work shows that Mif2 and OA can bind centromeric nucleosomes (*Kato et al., 2013*; *Xiao et al., 2017*; *Anedchenko et al., 2019*; *Fischböck-Halwachs et al., 2019*), which carry Cse4, a specialized variant of histone H3 (human CENP-A) (*Palmer et al., 1991*; *Stoler et al., 1995*). To ask if either Mif2 or OA can assemble in our assay onto these specialized nucleosomes, histone complexes containing H2A, H2B, H4, and Cse4, were wrapped in 601 DNA and the resulting nucleosome core particles (Cse4-NCPs) were bound to polystyrene microbeads. Mif2 or OA was then added in solution, along with 2D-MIND and Ndc80c. With OA in solution, 78% of beads bound microtubules; with Mif2 in solution, 24% bound (*Figure 5A*). Possible reasons for the relatively low percentage of active Cse4-NCP-decorated beads in the experiments with added Mif2/2D-MIND/Ndc80c are discussed below. Nevertheless, these observations confirm that OA and Mif2 can individually link the outer kinetochore subcomplexes, 2D-MIND/Ndc80c, to centromeric nucleosomes, spontaneously forming the four-component microtubule-binding chains, Cse4-NCPs/OA/2D-MIND/Ndc80c and Cse4-NCPs/Mif2/2D-MIND/Ndc80c, respectively. As expected (*Lang et al., 2018*), the Cse4-NCP-decorated beads completely failed to bind microtubules if both OA and Mif2 were omitted (*Figure 5A*), confirming that the NCPs themselves do not interact directly with MIND, Ndc80c, or microtubules. To test whether Mif2 and OA bind specifically to *centromeric* nucleosomes, we also wrapped canonical nucleosomes containing histone H3 (instead of Cse4) with 601 DNA and attached these H3-NCPs to beads. The H3-NCP-decorated beads did not bind microtubules with either Mif2 or OA in solution (*Figure 5A*), indicating that both Mif2 and OA bind selectively to centromeric,

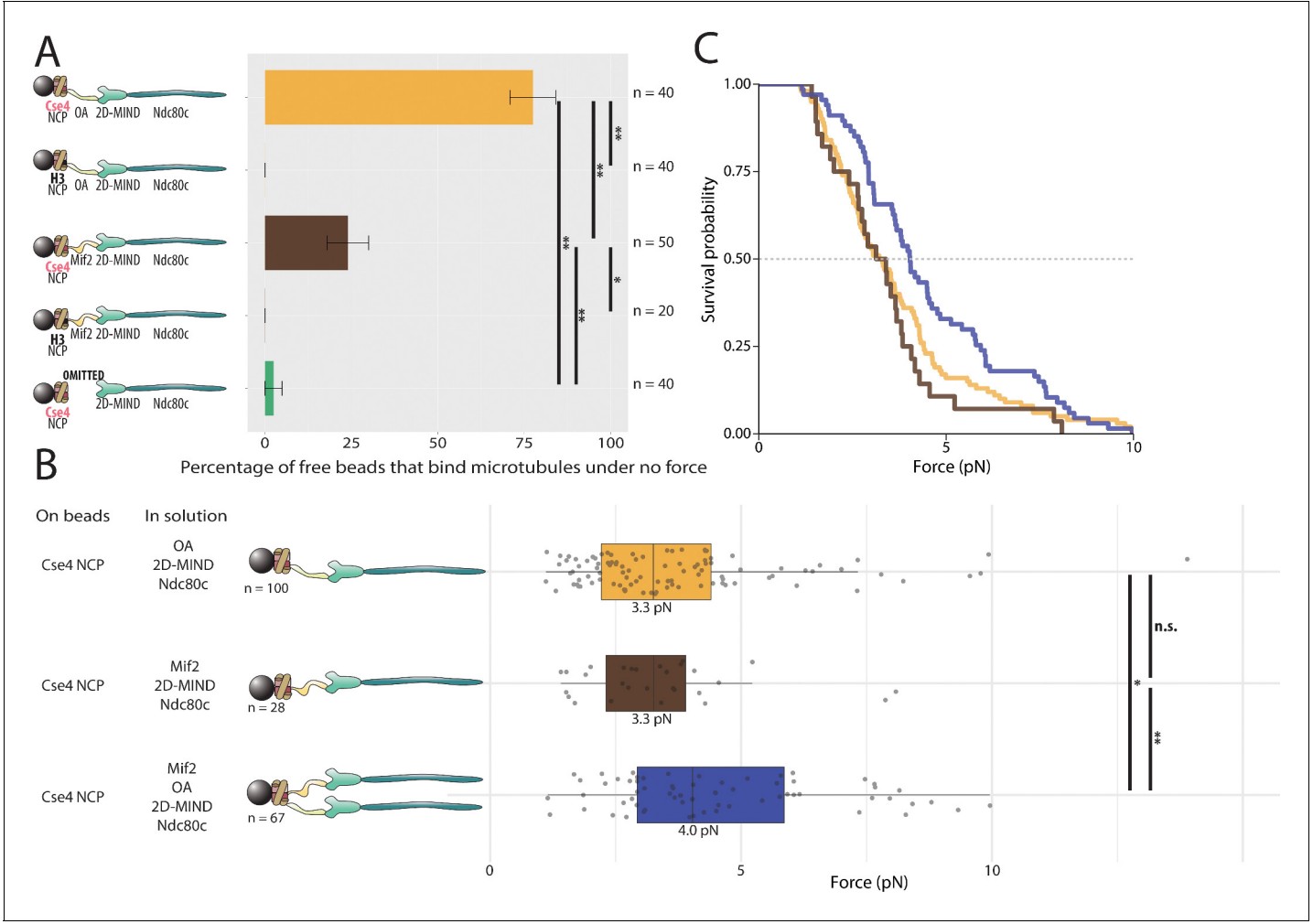

**Figure 5.** Assemblies based on centromeric nucleosomes form load-bearing microtubule attachments through OA or Mif2 or both. (**A**) Percentages of free beads that bound microtubules under no force. Error bars indicate the standard error of the sample proportion. Barnard's test was used to compare contingency tables. n.s. indicates p>0.05. * indicates p<0.05. ** indicates p<0.01. (**B**) Boxplot of rupture forces observed with reconstituted kinetochores. Each shaded circle is an individual rupture event. Boxes extend from the lower quartile to the upper quartile. Whiskers extend to 1.5 times the interquartile range beyond each quartile. A Kolmogorov-Smirnov test was used to compare probability distributions and calculate p-values. n.s. indicates p>0.05. * indicates p<0.05. ** indicates p<0.01. (**C**) Survival curves for Cse4 NCP/<u>OA</u>/2D-MIND/Ndc80c linkages (yellow), Cse4 NCP/Mif<u>2</u>/2D-MIND/Ndc80c linkages (brown), and Cse4 NCP/<u>OA/Mif2</u>/2D-MIND/Ndc80c linkages (purple). Raw data of all rupture events are included in *Figure 5—source data 1*.

The online version of this article includes the following source data and figure supplement(s) for figure 5:

**Source data 1.** Raw rupture force data for nucleosome core particle-based linkers.
**Figure supplement 1.** Load-bearing attachments between OA and Cse4-NCPs occur through the N-terminus of Cse4.
**Figure supplement 2.** Neither Dam1c nor CI increases the rupture force of NCP-containing linkers.

Cse4-containing nucleosomes. Altogether these results show that Mif2 and OA can serve as molecular bases for two distinct paths connecting the outer kinetochore to the centromere.

## Both OA and Mif2 form load-bearing attachments to centromeric nucleosomes

Having established that both OA and Mif2 can individually and selectively link bead-bound Cse4-NCPs to outer kinetochore subcomplexes, we used our force-ramp assay to test whether these interactions are load-bearing. With Cse4-NCPs on the beads, attachments mediated by Cse4-NCP/<u>Mif2</u>/2D-MIND/Ndc80c chains ruptured at 3.3 pN (*Figure 5B,C*). Likewise, attachments mediated by Cse4-NCP/<u>OA</u>/2D-MIND/Ndc80c chains also ruptured at 3.3 pN. The two survival curves are not

significantly different from one another (p=0.55) (*Figure 5B,C*). However, both are significantly weaker than the chains assembled with Mif2 or OA bound directly to the beads, without NCPs (p=0.00015 for the comparison with Mif2/2D-MIND/Ndc80c; p=0.00405 for OA/2D-MIND/Ndc80c).

Given our previous finding that OA strengthens Mif2-based chains (*Figure 4A,B*), we were curious whether the simultaneous presence of both inner kinetochore complexes, OA and Mif2, would strengthen Cse4-NCP-based chains. It did. With a median rupture force of 4.0 pN, Cse4-NCP-based chains with both Mif2 and OA added in solution were significantly stronger than chains containing only Mif2 (p=0.015) or containing only OA (p=0.037) (*Figure 5B,C*). This strengthening suggests that the Mif2/Cse4-NCP and OA/Cse4-NCP interfaces created in our assay are individually relatively weak but can mutually reinforce one another.

OA is reported to bind both DNA and the N-terminus of Cse4 (*Hornung et al., 2014*; *Anedchenko et al., 2019*; *Fischböck-Halwachs et al., 2019*). In order to assess the relative contributions of these two interactions to the OA/Cse4-NCP interface, we immobilized His$_6$MBP-tagged Cse4 (residues 1–50) on the surface of the beads and added free OA, 2D-MIND, and Ndc80c in solution (*Figure 5—figure supplement 1*). These Cse4$^{(1-50)}$-based chains exhibited a median rupture strength of 3.1 pN and were not significantly weaker than chains based on complete centromeric nucleosomes (p=0.91). This observation indicates that interactions of OA with the 601 DNA, the rest of Cse4, or histones other than Cse4 made no significant contribution to the strength of the OA/Cse4-NCP interface in our experiments. Because Mif2 homologs bind the C-terminus of Cse4 homologs (*Carroll et al., 2010*; *Guse et al., 2011*), we expected that adding Mif2 in solution would not strengthen the Cse4$^{(1-50)}$-based assemblies. Indeed, the addition of free Mif2 did not strengthen the Cse4$^{(1-50)}$/OA/2D-MIND/Ndc80c chains (p=0.33)(*Figure 5—figure supplement 1*).

To further assess the relative strength of the OA/Cse4-NCP interface, we added free Dam1c to the Cse4-NCP/OA/2D-MIND/Ndc80c chains (*Figure 5—figure supplement 2*). The addition of Dam1c, which specifically strengthens the Ndc80/microtubule interface, had no significant effect on the rupture strength of the Cse4-NCP/OA/2D-MIND/Ndc80c chains (p>0.5), indicating that the Cse4-NCP/OA interface is probably the weakest link in these chains. We attempted to strengthen that interface by adding Chl4/Iml3 (CI) in solution, but it also had no effect on overall chain strength (p>0.5) (*Figure 5—figure supplement 2*). Altogether, these data indicate that both essential components of the inner kinetochore, OA and Mif2, are capable of independently transmitting piconewton-scale forces from the outer kinetochore to centromeric nucleosomes.

## Discussion

We have reconstituted kinetochore assemblies capable of transmitting piconewton-scale forces from dynamic microtubules to Cse4-containing nucleosomes (*Figure 5*). To our knowledge, these are the first kinetochore assemblies made from individually purified, recombinant *S. cerevisiae* components that reconstitute what is arguably the kinetochore's most fundamental activity: coupling centromeric nucleosomes to dynamic microtubule tips under physiologically relevant loads.

The self-assembly of full-length kinetochore subcomplexes at nanomolar concentrations suggests that they might have greater affinities for one another than suggested by the micromolar dissociation constants obtained using short peptides (*Hornung et al., 2014*; *Dimitrova et al., 2016*; *Petrovic et al., 2016*), possibly because of additional points of contact, although the immobilization of one subcomplex on beads might also affect binding dynamics. Both OA and Mif2 are individually capable of forming load-bearing interactions with MIND, demonstrating that there are two distinct paths of force transmission through the inner kinetochore. However, these paths are not equivalent. They are differentially sensitive to regulation by the major mitotic kinase, Ipl1 (Aurora B in humans), with the Mif2 path depending on the presence of phosphomimetic mutations on the MIND complex, whereas the OA pathway is only slightly strengthened by them. Also, while both the Mif2/2D-MIND and OA/2D-MIND interfaces are load-bearing, the former is mechanically weak relative to the Ndc80c/microtubule interface, but the latter is strong (*Figure 2* and *Figure 3*).

We also found that OA can strengthen Mif2/2D-MIND/Ndc80c linkers, whereas Mif2 slightly weakens OA/2D-MIND/Ndc80c linkers. Considering that '2D' phosphomimetic mutations on MIND partially alleviate autoinhibition but do not abrogate it completely (*Dimitrova et al., 2016*), we speculate that OA binding to 2D-MIND might alleviate this residual autoinhibition, promoting MIND's interaction with Mif2 and strengthening the Mif2-based chain. In contrast, for the OA-based linker,

we hypothesize that OA disinhibition of 2D-MIND might permit Mif2 to outcompete OA for binding to 2D-MIND (*Figure 4A,B*). In other words, Mif2 might decrease the rupture force of the OA/2D-MIND/Ndc80c linkers by 'stealing' the disinhibited 2D-MIND from bead-bound OA, thus detaching MIND from the bead. Competition for binding to MIND is plausible, because previous work has shown that OA and Mif2 bind to adjacent sites on the same head of MIND (*Dimitrova et al., 2016*).

When NCPs are on the beads, adding both OA and Mif2 in solution strengthens their attachment to microtubules (*Figure 5B,C*). OA and Mif2 probably do not compete with each other for binding to a centromeric nucleosome, because OA binds the N-terminus of Cse4, whereas Mif2 likely binds the C-terminus of Cse4 (*Carroll et al., 2010*; *Guse et al., 2011*; *Anedchenko et al., 2019*; *Fischböck-Halwachs et al., 2019*). If OA and Mif2 each recruit one MIND complex to the nucleosome, then together they would increase the total number of Ndc80c complexes in the assembly. Both OA and Mif2 self-assemble exclusively on centromeric nucleosomes (*Figure 5A*), a kinetochore function essential for genome stability. Yet we consistently find that Cse4-NCP-based chains are weak compared to shorter chains without NCPs. We hypothesize that this relative weakness occurs because the Cse4-NCPs were wrapped in 601 DNA rather than centromeric DNA. Budding yeast Mif2 is reported to have a 40-fold greater affinity for Cse4-containing nucleosomes if they are wrapped in centromeric DNA (*Xiao et al., 2017*). We were unable to test the effect of this increased affinity on rupture force because wrapping nucleosomes in centromeric DNA proved intractable. Nonetheless, the rupture strengths of our reconstituted assemblies fall within the range of mitotic forces estimated for budding yeast kinetochores in vivo.

In addition to the absence of centromeric DNA, there are other possible reasons why the strongest chains studied here still had a median force below the 9.1 pN strength measured previously for native purified kinetochore particles (*Akiyoshi et al., 2010*). First, studies of native kinetochore particles have not tested force transmission through the inner kinetochore; force has been exerted only through outer kinetochore components via MIND-based bead linkages (*Akiyoshi et al., 2010*). Thus, the inner kinetochore connections might also be weaker in the native particles. Second, post-translational modifications retained on the native proteins but absent from our recombinant proteins might strengthen interactions within the inner kinetochore. Given that methylation of Cse4 is known to *weaken* the OA-Cse4 interface (*Anedchenko et al., 2019*), it seems plausible that additional modifications that *strengthen* the inner kinetochore might also be found.

Although both OA and Mif2 can transmit force from Cse4-NCPs to MIND, they are neither interchangeable nor redundant. There are several reasons that having multiple paths of force transmission might be advantageous. First, each path can be differentially regulated, which our data suggest is the case. Second, OA and Mif2 together increase the total number of outer kinetochore components that can be recruited. Estimates of the number of Ndc80 complexes present in each kinetochore vary from six to thirty copies (*Joglekar et al., 2006*; *Lawrimore et al., 2011*; *Dhatchinamoorthy et al., 2017*), and biophysical experiments demonstrate that multiple Ndc80 complexes are necessary for effective microtubule coupling (*Powers et al., 2009*; *Volkov et al., 2018*). But MIND and Ndc80c interact in a 1:1 ratio (*Dimitrova et al., 2016*), implying that the inner kinetochore must serve as an oligomerization platform, assembling on a single centromeric nucleosome and recruiting multiple copies of MIND and Ndc80c. If one OA complex and one Mif2 dimer assembled onto each of the two faces of a centromeric nucleosome, they could collectively recruit up to 6 MIND/Ndc80c, with additional copies of Ndc80c potentially being recruited by the Cnn1 pathway (*Lang et al., 2018*).

Our findings also raise intriguing questions about how these multiple possible paths of force transmission and Ndc80c recruitment may be differentially utilized at different stages of the cell cycle or by different organisms. For instance, the human homolog of OA binds microtubules (*Pesenti et al., 2018*), but budding yeast OA clearly does not (*Figure 2A*). For some organisms, OA is essential (*Pot et al., 2005*; *Kagawa et al., 2014*); other species appear to lack OA homologs (*Liu et al., 2016*; *van Hooff et al., 2017*). A smaller subset of organisms has no known homolog of Mif2 (*Drinnenberg et al., 2014*; *van Hooff et al., 2017*). Evidently, the relative importance of the OA and Mif2 pathways varies between organisms. In budding yeast, the extreme N-terminus of Ame1 is essential, while the N-terminus of Mif2 is not (*Hornung et al., 2014*), suggesting the primacy of the OA-based path in this organism. There is also strong evidence for a third pathway of Ndc80c recruitment, through Cnn1 (*Bock et al., 2012*; *Huis In 't Veld et al., 2016*; *Lang et al., 2018*). Although Cnn1 is not essential in yeast (*Giaever et al., 2002*; *De Wulf et al., 2003*), an

additive growth defect is observed when deletion of Cnn1 is combined with deletion of the N-terminus of Mif2 (*Hornung et al., 2014*). Homologs of Cnn1 also play important roles in several other organisms (*Gascoigne et al., 2011*; *Nishino et al., 2013*; *Huis In 't Veld et al., 2016*; *Cortes-Silva et al., 2020*). Thus, while the branched architecture of the inner kinetochore is a feature conserved across many species, the relative importance of particular branches apparently varies between species.

Taken together, our work represents a major advance towards the goal of reconstituting a complete, functional kinetochore from purely recombinant proteins. Going forward, the reconstituted 'skeleton' kinetochore presented here can be used as a scaffold to study the contributions of other kinetochore proteins to force transmission. Of particular interest are Cnn1, which could increase rupture force by recruiting additional copies of Ndc80c (*Schleiffer et al., 2012*; *Lang et al., 2018*), and Stu2, which is known to contribute significantly to kinetochore-microtubule coupling (*Miller et al., 2016*).

# Materials and methods

**Key resources table**

| Reagent type (species) or resource | Designation | Source or reference | Identifiers | Additional information |
|---|---|---|---|---|
| Strain, strain background (*Escherichia coli*) | Rosetta (DE3) pLys competent cells | Novagen | Cat#71403 | |
| Strain, strain background (*Escherichia coli*) | Rosetta (DE3) competent cells | Millipore Sigma | Cat#70954 | |
| Biological sample (*Bos taurus*) | Bovine brain tubulin | Lab purification | | Protocol adopted from *Castoldi and Popov, 2003* |
| Antibody | Penta-HIS biotin conjugate, monoclonal mouse | Qiagen | Cat#34440 | |
| Chemical compound, drug | Glucose oxidase | Millipore Sigma | Cat#345386 | |
| Chemical compound, drug | Catalase | Millipore Sigma | Cat#219261 | |
| Chemical compound, drug | Biotinylated bovine serum albumin (BSA) | Vector laboratories | Cat#B-2007 | |
| Chemical compound, drug | Avidin DN | Vector laboratories | Cat#A-3100 | |
| Chemical compound, drug | TCEP | Thermo Fischer | Cat#T2556 | |
| Software, algorithm | Labview | National Instruments | RRID:SCR_014325 | |
| Software, algorithm | Igor Pro | Wavemetrics | RRID:SCR_000325 | |
| Software, algorithm | R | R Foundation for Statistical Computing | | |

## Plasmids and constructs

All expression vectors are listed in *Figure 1—source data 1*.

## Protein expression and purification

### Dam1c, Ndc80c, and MIND

Dam1c, MIND, and all MIND mutants were expressed in *Escherichia coli* from polycistronic vectors. Ndc80c was expressed from two bicistronic vectors encoding Ndc80/Nuf2 and Spc24/Spc25 (*Wei et al., 2005*). The protein subcomplexes purified as previously described using either a $His_6$- or FLAG- affinity tag followed by size exclusion chromatography (SEC)(*Gestaut et al., 2008*; *Gestaut et al., 2010*; *Kudalkar et al., 2015*).

Briefly, His$_6$-tagged protein subcomplexes were expressed from polycistronic pST39 vectors in BL21 cells and induced with 0.4 mM isopropyl β-D-1-thiogalactopyranoside (IPTG) for 16 hr at 18˚C. Cells were lysed with a French press and the subcomplex was immobilized on a Ni-charged IMAC resin column (Bio-Rad) in either MIND buffer (50 mM sodium phosphate buffer, pH 7.0, 200 mM NaCl), Ndc80c buffer (50 mM HEPES buffer, pH 7.6, 300 mM NaCl), or Dam1c buffer (50 mM sodium phosphate buffer, pH 6.9, 500 mM NaCl) supplemented with protease inhibitors (Roche), 5 mM imidazole, and 1 mM PMSF. The resin was then washed, and subcomplex eluted with buffer containing 300 mM imidazole before being further purified using a Superdex 200 size exclusion column (GE Healthcare) in either MIND buffer, Ndc80c buffer with 200 mM NaCl, or Dam1c buffer.

## Okp1/Ame (OA)

Wild-type (WT) and truncated His$_6$-tagged OA was expressed in a polycistronic pST39 vector in BL21 cells and induced with 0.4 mM IPTG for 16 hr at 18˚C. Cells were lysed with a French press and OA was immobilized on a Ni-charged IMAC resin column (Bio-Rad) in 50 mM sodium phosphate buffer, pH 7.0 supplemented with 200 mM NaCl, protease inhibitors (Roche), 5 mM imidazole, and 1 mM PMSF, washed, and eluted in the same buffer with 300 mM imidazole. OA was further purified using a Superdex 200 size exclusion column (GE Healthcare) in SEC-OA buffer (50 mM sodium phosphate buffer, pH 7.0 supplemented with 200 mM NaCl.)

WT and truncated FLAG-tagged OA was expressed similarly to His$_6$-tagged OA except that it was immobilized on an anti-FLAG M2 affinity gel (A2220; Sigma). The affinity gel was washed, and protein was eluted with 0.1 mg·mL$^{-1}$ 3X FLAG Peptide (F4799; Sigma). OA was further purified using a Superdex 200 size exclusion column (GE Healthcare) in SEC-OA buffer.

After SEC, OA was concentrated with Amicon Ultra centrifugal filters (Millipore), stored in SEC-OA buffer with 5% glycerol, and immediately stored at −80˚C.

## Mif2

Mif2 expression plasmids contain a C-terminal (Ct) linker and maltose-binding protein (MBP) tag in order to improve solubility and expression of Mif2. The location of the tag was rationally selected, as the N-terminus of Mif2 homologs is reported to mediate binding to MIND (*Dimitrova et al., 2016*; *Petrovic et al., 2016*), while a central domain mediates binding to the centromeric nucleosome (*Carroll et al., 2010*; *Kato et al., 2013*).

Full length Mif2- (Ct)MBP tag and Mif2-(Ct)MBP-His$_6$ constructs were expressed in Rosetta 2 DE3 pLysS competent cells (Novagen) and expression was induced with 0.25 mM IPTG at 18˚C for 16 hr. Cells were lysed with either sonication or a French press. Mif2-(Ct)MBP was immobilized on amylose resin (NEB) in Mf1-buffer (30 mM HEPES buffer, pH 7.5, 2 M NaCl, 10% glycerol, 1 mM TCEP, protease inhibitors (Roche) and 1 mM PMSF). The column was washed with Mf1-buffer and the protein eluted with Mf1-buffer containing 10 mM maltose. Protein samples were immediately loaded on HiTrap Q HP (GE Healthcare) anion exchange column in buffer QA (30 mM HEPES buffer, pH 7.5, 50 mM NaCl, 1 mM TCEP, 10% glycerol) and eluted with a 0–100% gradient of buffer QB (30 mM HEPES buffer, pH 7.5, 1 M NaCl, 1 mM TCEP, 10% glycerol). This was followed by SEC on a preparative Superdex200 HiLoad 16/60 column (GE Healthcare) in SEC-Mif2 buffer (30 mM HEPES buffer, pH 7.5, 200 mM NaCl, 1 mM TCEP). Purification of all Mif2 constructs were performed at 4˚C to prevent protein degradation. Mif2 protein was immediately concentrated, frozen, and stored at −80˚C.

## Chl4/Iml3 (CI)

CI was expressed from a bicistronic vector (derived from pSMH104) in Rosetta DE3 pLysS competent cells (Novagen) and expression induced with 0.25 mM IPTG as previously described (*Hinshaw and Harrison, 2013*). Cells were lysed with a French press. CI was immobilized on anti-FLAG M2 affinity resin (Sigma) in CI-buffer of 50 mM HEPES buffer, pH 7.5, 100 mM NaCl, 1 mM EDTA, 10% glycerol, protease inhibitors (Roche) and 1 mM PMSF. The affinity gel was washed, and protein was eluted with SEC-CI buffer (50 mM Tris buffer, pH 8.5, 200 mM NaCl, 1 mM TCEP) supplemented with 0.1 mg·mL$^{-1}$ 3X FLAG Peptide (F4799; Sigma). CI was further purified using a Superdex 200 size exclusion column (GE Healthcare) in SEC-CI buffer (50 mM Tris-Cl, pH 8.5, 200 mM NaCl and 1 mM TCEP). After SEC, all proteins were concentrated with Amicon Ultra centrifugal filters (Millipore),

stored in SEC-CI buffer from the final Superdex 200 run with 5% glycerol, and immediately stored at −80°C.

### Cse4$^{(1-50)}$

The first 50 residues of Cse4 were expressed with a TEV-cleavable His$_6$MBP-tag in BL21(DE3) cells. Upon reaching OD$_{600}$ of 0.6–1.0, expression was induced with 0.25 mM IPTG for 16 hr at 18°C. Cells were lysed by sonication and Cse4$^{(1-50)}$ was immobilized on HisPur cobalt resin (ThermoFisher) in 30 mM HEPES buffer, pH 7.5 supplemented with 600 mM NaCl, 10 mM imidazole, protease inhibitors (Roche), 1 mM TCEP, 10% glycerol, and 1 mM PMSF. The resin was washed with the same buffer and protein was eluted with buffer QA (30 mM HEPES buffer, pH 7.5, 50 mM NaCl, 1 mM TCEP, 10% glycerol) containing 600 mM imidazole. Protein samples were immediately loaded on HiTrap Q HP (GE Healthcare) anion exchange column in buffer QA and eluted with a 0–100% gradient of buffer QB (30 mM HEPES buffer, pH 7.5, 1 M NaCl, 1 mM TCEP, 10% glycerol). Cse4$^{(1-50)}$ was further purified using a Superdex 200 size exclusion column (GE Healthcare) in 30 mM HEPES buffer, pH 7.5 supplemented with 150 mM NaCl and 1 mM TCEP. After SEC, Cse4$^{(1-50)}$ was concentrated with Amicon Ultra centrifugal filters (Millipore) in buffer from the final Superdex 200 run with 5% glycerol and immediately stored at −80°C.

### Nucleosome core particles (NCPs)

Each histone complex was co-expressed from a single expression polycistronic vector encoding *S. cerevisiae (Sc)* and *Kluyveromyces lactis (Kl)* genes: pScKl2 containing His$_6$-*Kl*-H2A / His$_6$-*Kl*-H2B/ *Sc*-Cse4 / His$_6$-*Kl*-H4 and pScKl4 containing *Sc*-H3 / His$_6$-*Sc*-H2A/*Sc*-H2B / His$_6$-*Kl*-H4 (*Migl et al., 2020*). Histones were co-expressed from pScKl2 or pScKl4 in Rosetta 2 DE3 pLysS competent cells (Novagen) and upon reaching OD$_{600}$ of 0.6–1.0, expression was induced with 0.25–0.5 mM IPTG and cells grown for 16 hr at 18°C. Cells were resuspended in High Salt Buffer (HSB) (2 M NaCl, 50 mM HEPES buffer, pH 7.5, 10% glycerol, 1 mM TCEP and protease inhibitors: aprotinin, pepstatin, leupeptin, and PSMF), frozen and stored at –80°C. Cells were lysed by sonication and lysate clarified by centrifugation at 40,000xg at 4°C for 1 hr. Histones were immobilized on TALON metal affinity resin (Clontech) and incubated at 4°C for 1 hr with agitation, then transferred to a gravity column.

The flowthrough was discarded, beads washed with HSB, and histones eluted with 400 mM imidazole and 50 mM EDTA. The eluate was concentrated, and histone octamers were further purified on a Superdex 200 size exclusion column (GE Healthcare) in SEC-H1 buffer containing 2M NaCl, 50 mM HEPES buffer, pH 7.5, 1 mM TCEP.

Histone octamers were wrapped in the 147-basepair double-stranded Widom DNA (henceforth '601 DNA') (*Lowary and Widom, 1998*) as described (*Migl et al., 2020*). 601 DNA was produced by PCR and final purification performed using HiTrapQ column. 601 DNA and histone octomers were combined in a 1.1:1 molar ratio in a hydrated 7K MWCO Slide-A-Lyzer MINI Dialysis Device (69560; ThermoFisher), which was placed in 0.5 L of high-salt dialysis buffer (2 M NaCl, 30 mM HEPES buffer, pH 7.5, 1 mM TCEP, and 5 mM EDTA). The nucleosome wrapping was achieved through dialysis against low salt dialysis buffer (200 mM NaCl, 30 mM HEPES buffer, pH 7.5, 1 mM TCEP, and 5 mM EDTA) over the course of 60 hr at 4°C. Purification of excess DNA was done by using a Superdex 200 or Superose six size exclusion columns (GE Healthcare) with SEC-H2 buffer 150 mM NaCl, 30 mM HEPES buffer, pH 7.5, 1 mM TCEP. NCPs were stored on ice, and all experiments using NCPs were performed within one week of wrapping.

## Optical trap rupture force assay

Streptavidin-coated 0.56 µm polystyrene beads (Spherotech) were coated with biotinylated anti-His5 antibodies (Qiagen), and 3.5 pM beads were incubated with 10 nM His$_6$-tagged protein complex as described (*Franck et al., 2010*; *Umbreit et al., 2014*), such that each bead was decorated with ~3000 protein complexes. Flow cells were prepared using double-sided tape and plasma-cleaned coverslips and incubated with 25 µL of 1 mg·mL$^{-1}$ biotinylated BSA (Vector Laboratories) for 15 min at room temperature, then washed with BRB80 (80 mM PIPES, pH 6.9, 1 mM MgCl$_2$, 1 mM EGTA). Flow cells were then incubated with 30 µL of 1 mg·mL$^{-1}$ avidin DN (Vector Laboratories) for 5 min at room temperature, followed by another BRB80 wash. GMPCPP biotinylated tubulin seeds in BRB80 were bound for 5 min and washed twice with 42°C growth buffer (BRB80 plus 1 mM GTP and 105

μM κ-casein). Protein-decorated beads were introduced into the flow chamber in growth buffer with 1.4 mg·mL$^{-1}$ tubulin, 313 μg·mL$^{-1}$ glucose oxidase, 37.5 μg·mL$^{-1}$ catalase, 30 mM glucose, 1 mM DTT, and 10 nM each of the free, non-His-tagged protein complexes.

For the negative controls (e.g., OA-decorated beads with Ndc80c but not MIND in solution), all beads and soluble proteins were maintained at the concentrations used in the experimental reactions, except for the subcomplex being omitted.

For each condition, a bead was manipulated to contact a single microtubule (*Figure 1—figure supplement 1*). If the bead-microtubule contact did not persist after the laser was switched off, that bead was considered to be a nonbinder. Optical trap assays were performed at room temperature using custom instrumentation to capture and manipulate beads as described (*Franck et al., 2010*). Rupture force assays were performed as described (*Akiyoshi et al., 2010*; *Tien et al., 2010*; *Sarangapani et al., 2013*). Once beads were bound to microtubule tips, a test force of 1 pN was applied, and only beads that tracked with ~100 nm of tip growth were subjected to ramping force of 0.25 pN·s$^{-1}$ until detachment. All attachments that withstood the 1 pN preload force were included in our analysis. Records of force vs. time were collected, and maximum rupture force was determined using custom Igor Pro software (Wavemetrics) (Source data 1). Protein chains that withstood the 1-pN preload force were considered load-bearing, and distributions of rupture force were used to compare relative mechanical strength.

## Data analysis and figure preparation

Data from optical trap assays were analyzed in R (*Wickham, 2009*; *R Development Core Team, 2013*; *Kassambra and Kosiniski, 2018*) and Igor Pro (Wavemetrics). Figures were produced using R and Adobe Illustrator.

## Acknowledgements

We thank Stephen Hinshaw (Harrison Lab, HMS) for a generous gift of Chl4/Iml3 expression vector pSMH104 and David Migl (Harrison Lab, HMS) for a generous gift of purified H3-H2A-H2B-H4 histone octamer. We also thank Michael Riffle for assistance with data analysis and visualization.

The research was supported by NIH grants R01 GM040506 and R35 GM130293 (TND), R01 GM079373 and R35 GM134842 (CLA), and NIH Training Grant in Molecular Biophysics T32 GM008268 (GEH).

YND and CLN are employees at Genentech, Inc.

## Additional information

### Competing interests

Cameron L Noland, Yoana N Dimitrova: is affiliated with Genentech Inc. The author has no financial interests to declare. The other authors declare that no competing interests exist.

### Funding

| Funder | Grant reference number | Author |
|---|---|---|
| National Institutes of Health | Training Grant in Molecular Biophysics T32 GM008268 | Grace Elizabeth Hamilton |
| National Institutes of Health | R01 GM040506 | Trisha N Davis |
| National Institutes of Health | R35 GM130293 | Trisha N Davis |
| National Institutes of Health | R01 GM079373 | Charles L Asbury |
| National Institutes of Health | R35 GM134842 | Charles L Asbury |
| Genentech | | Cameron L Noland Yoana N Dimitrova |

The funders had no role in study design, data collection and interpretation, or the decision to submit the work for publication.

## Author contributions

Grace E Hamilton, Conceptualization, Formal analysis, Funding acquisition, Validation, Investigation, Visualization, Methodology, Writing - original draft, Writing - review and editing; Luke A Helgeson, Resources, Investigation, Methodology, Writing - review and editing; Cameron L Noland, Resources, Histone purification and nucleosome core particle wrapping; Charles L Asbury, Conceptualization, Resources, Software, Formal analysis, Supervision, Funding acquisition, Methodology, Writing - review and editing; Yoana N Dimitrova, Conceptualization, Resources, Formal analysis, Supervision, Investigation, Project administration, Writing - review and editing; Trisha N Davis, Conceptualization, Supervision, Funding acquisition, Methodology, Project administration, Writing - review and editing

## Author ORCIDs

Grace E Hamilton ⓘ https://orcid.org/0000-0002-0522-0702
Luke A Helgeson ⓘ http://orcid.org/0000-0001-5112-2751
Cameron L Noland ⓘ https://orcid.org/0000-0001-6364-3167
Charles L Asbury ⓘ https://orcid.org/0000-0002-0143-5394
Yoana N Dimitrova ⓘ https://orcid.org/0000-0003-1547-5781
Trisha N Davis ⓘ https://orcid.org/0000-0003-4797-3152

## Decision letter and Author response

Decision letter https://doi.org/10.7554/eLife.56582.sa1
Author response https://doi.org/10.7554/eLife.56582.sa2

# Additional files

## Supplementary files

• Supplementary file 1. Proteins of the kinetochore and plasmids used in this study. Colour versions of *Tables 1* and *2*.

• Transparent reporting form

## Data availability

All data analyses during this study are included in the manuscript and supporting files. Source data files have been provided for Figures 1-5 and their supplements.

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
