## [Decision Letter]

**Acceptance summary:**

The kinetochore is a fascinating large protein assembly on eukaryotic chromosomes. When chromosomes need to be moved during cell division, microtubules attach to kinetochores and pull the chromosomes into the daughter cells. The kinetochore must withstand these pulling forces and transmit the force from microtubules to the chromosomal DNA. How each of the many kinetochore proteins is involved in transmitting force has been unclear. In this study, Hamilton and colleagues have reconstituted parts of the budding yeast kinetochore and have used a force rupture assay to test what forces these assemblies can withstand. Their findings uncover at least two possible paths of force transmission through the kinetochore. This is an important complement to recent advances in revealing the structure of kinetochores, and the findings provide crucial information for future research into how kinetochores function at different stages of cell division and in different organisms.

**Decision letter after peer review:**

Thank you for submitting your article "Reconstitution reveals two paths of force transmission through the kinetochore" for consideration by *eLife*. Your article has been reviewed by three peer reviewers, one of whom is a member of our Board of Reviewing Editors, and the evaluation has been overseen by a Reviewing Editor and Anna Akhmanova as the Senior Editor. The following individual involved in review of your submission has agreed to reveal their identity: Stefan Westermann (Reviewer #3).

The reviewers have discussed the reviews with one another and the Reviewing Editor has drafted this decision to help you prepare a revised submission.

As the editors have judged that your manuscript is of interest, but as described below, additional experiments are required before it is published. We would like to draw your attention to changes in our revision policy that we have made in response to COVID-19 (https://elifesciences.org/articles/57162). First, because many researchers have temporarily lost access to the labs, we will give authors as much time as they need to submit revised manuscripts. We are also offering, if you choose, to post the manuscript to bioRxiv (if it is not already there) along with this decision letter and a formal designation that the manuscript is 'in revision at *eLife*'. Please let us know if you would like to pursue this option. (If your work is more suitable for medRxiv, you will need to post the preprint yourself, as the mechanisms for us to do so are still in development.)

Summary:

The kinetochore is a fascinating molecular machine in eukaryotes, which transmits force from microtubules to the centromeric DNA in order to allow for accurate chromosome segregation. How each kinetochore protein is involved in force transmission is still unclear. In this study, Hamilton and colleagues have reconstituted parts of the budding yeast kinetochore and have used a force rupture assay to test what forces these assemblies can withstand. The results provide insight into how force might be transmitted between the microtubule attachment site of the kinetochore and the centromeric DNA. This complements recent advances in understanding the structure and assembly of kinetochores.

Hamilton and colleagues have immobilized parts of yeast kinetochores (assembled from recombinant subcomplexes) on beads and have tested their interaction with dynamic microtubule ends in an optical trap assay. This type of assay has been introduced to the field by the Asbury lab, and has been mostly used for natively purified kinetochores, or individual recombinant components of the outer kinetochore such as Dam1 or Ndc80. Here, more complex recombinant reconstitutions, involving the centromeric nucleosome and the essential CCAN components Mif2(CENP-C) and OA(CENP-Q/U) are tested for their ability to form load-bearing links to microtubule ends.

There are some conceptual limitations to the experimental setup which make it not straightforward to relate the results to kinetochores inside cells: (1) the size of the bead and the immobilization of nucleosomes on the bead surface likely reflect the real chromatin environment only poorly; (2) perhaps more importantly, the actual molecular nature of the link that is assembled on the bead surface and interacts with the microtubule cannot be monitored directly (for example, how many Ndc80 molecules are recruited in each case?), but is only inferred based on the result of the experiment; (3) while the assays can probe individual interactions, these interactions may be modified in vivo (e.g. by additional kinetochore proteins or post-translational modifications), so that the assay only provides a partial picture.

With these limitations in mind, the approach of using reconstituted components to reveal properties of distinct molecular linkages within the kinetochore is informative. The authors have found that the connection between CENP-QU(=OA) and MIND is stronger in their assay and less dependent on MIND phosphorylation than the one between CENP-C(=Mif2) and MIND. This finding complements the prior information that disrupting Mif2-MIND interaction by Mif2 N-terminal deletion has much less drastic consequences on cells than disrupting OA-MIND interaction by Ame1 N-terminal deletion.

While the authors find some strengthening of the centromere/microtubule connection when both Mif2 and OA are present, the interplay between these two important elements remains poorly explored. The reviewers find it necessary that the authors use well-characterized budding yeast kinetochore mutants in their assay to dissect different possibilities.

Essential revisions:

1) In the assays that contain both Mif2 and OA (Figure 3B,C and 4B,C), it will be important to establish whether increases in rupture force are due to a strengthening of the Mif2-MIND interaction by OA or due to additional OA-MIND-Ndc80 linkages (increased valency). This can be addressed with the ΔN-OA mutant that the authors have available (Figure 2). Does ΔN-OA increase the rupture force with either Mif2 or Cse4-NCP on the bead? Alternatively, other published mutants (e.g. MIND point mutants) could serve a similar purpose. In the assay with Mif2 on beads (+ OA, 2D-MIND, Ndc80c), the authors should also add Dam1c to further probe the strength of this linkage.

Since the contacts of OA and Mif2 with Cse4 seem distinct, N- or C-terminal truncations of Cse4 could address how much of the strengthening comes from direct OA-Mif2 interactions vs. full NCP-OA/Mif2-MIND-Ndc80 linkages, as shown in the cartoon in Figure 4B.

2) The authors may also want to re-think how they discuss their results in the background of the published yeast genetics data. While simultaneously adding Mif2 and OA to the Cse4 NCP gives increased strength of the attachment, the relation of this finding to native kinetochores is not straightforward: ΔN-Mif2 (defective in MIND binding) is viable, suggesting that the Mif2 based link of "force transmission" is not essential in yeast. Maybe the concept of different "paths of force transmission" through the kinetochore is not helpful at this point. Maybe distinct "load-bearing molecular links" within the kinetochore would be a better term. If the authors decide to revise this, they also need to change the title of the paper.

3) The discussion of what exactly the authors' assay measures should be improved. It was brought up whether it is indeed "mechanical strength" of a non-covalent bond that is measured in this assay. This would only be the case if the half-life of the respective bond was significantly longer than the time scale of the experiment (otherwise the assay is measuring the probability of the dissociation of the assembled complex in equilibrium with soluble components, determined by the thermodynamic dissociation rate constant). Is this condition met in the assay? The concentration of soluble subcomplexes is kept at 10 nM throughout. Isn't this an important experimental variable? Have the authors tried to vary the concentration? What is the effect in the binding and rupture assays?

4) The authors should double-check their citations to make sure they are appropriate. Some problems that the reviewers identified were:

- Subsection “OA forms microtubule attachments through MIND and Ndc80c”: The sentence deals with Ame1-MIND binding, but the cited Screpanti paper is about human CENP-C – Mis12 complex interaction.

- Subsection “Mif2 forms microtubule attachments through MIND and Ndc80c”: The authors want to argue that parts of Mis2 other than the very N-terminus may contribute to MIND binding and cite Cheeseman et al., 2002. This is not covered in this publication, and the reviewers are not aware of other evidence.

- Discussion section: The sentence is about the different binding mode of OA and Mif2 to the centromeric Cse4-containing nucleosome, but the Straight lab papers that are cited deal with binding of vertebrate CENP-N and CENP-C to CENP-A containing nucleosomes, which does not directly support the argument.

- Subsection “Mif2 forms microtubule attachments through MIND and Ndc80c”: The binding interface is with Mtw1-Nnf1 (MN, head I), not with Dsn1-Nsl1. Please correct.

- Discussion section: Yan et al., 2019, could be cited when discussing the weak interactions when using Cse4 NCPs assembled on 601 DNA and the potential role of centromeric DNA.

- Fischboeck et al. is still cited as Fischboeck et al., 2018 (bioRxiv), but should be Fischboeck et al., 2019 (*eLife*).

---

## [Author Response]

Essential revisions:1) In the assays that contain both Mif2 and OA (Figure 3B,C and 4B,C), it will be important to establish whether increases in rupture force are due to a strengthening of the Mif2-MIND interaction by OA or due to additional OA-MIND-Ndc80 linkages (increased valency). This can be addressed with the ΔN-OA mutant that the authors have available (Figure 2). Does ΔN-OA increase the rupture force with either Mif2 or Cse4-NCP on the bead? Alternatively, other published mutants (e.g. MIND point mutants) could serve a similar purpose.

We agree with the reviewers that this question of how OA increases the strength of Mif2-based chains is very interesting. As requested, we have added new data demonstrating that the addition of OAdeltaN does not increase the rupture strength of Mif2/2D-MIND/Ndc80c linkers. Unfortunately, interpretation of this result is not simple. While removing the N-terminus of Ame1 precludes the possibility that OA is recruiting additional copies of MIND-Ndc80c, it is also possible that deleting the N-terminus of Ame1 abrogates OA’s ability to alleviate autoinhibition within the MIND complex. Thus, these new results are consistent with either the increased valency or the alleviation of autoinhibition hypotheses. Nevertheless, we agree that they are interesting, and we have added them to the revised manuscript (in what is now Figure 4 and described in subsection “OA strengthens the Mif2/2D-MIND interface” of the text).

In the assay with Mif2 on beads (+ OA, 2D-MIND, Ndc80c), the authors should also add Dam1c to further probe the strength of this linkage.

We thank the reviewers for this excellent suggestion. At their request, we have added the following new experiment to what is now Figure 4: Mif2/OA/2D-MIND/Ndc80c/Dam1c. The addition of Dam1c to the Mif2/OA/2D-MIND/Ndc80c linker did not increase its rupture force. This new finding provides strong, additional support for our conclusion that the Mif2/2D-MIND interface remains a “weak link” in the Mif2based chains, even in the presence of OA and phosphomimetic mutations to MIND. (See Figure 4 and subsection “OA strengthens the Mif2/2D-MIND interface”.)

Since the contacts of OA and Mif2 with Cse4 seem distinct, N- or C-terminal truncations of Cse4 could address how much of the strengthening comes from direct OA-Mif2 interactions vs. full NCP-OA/Mif2-MIND-Ndc80 linkages, as shown in the cartoon in Figure 4B.

We agree that truncations of Cse4 can help dissect which interactions are sufficient for the observed strengthening. In the revised manuscript we have added new data showing that the mechanical strength of Cse4-NCP-based chains with free OA added in solution is comparable to the strength of chains based on the isolated N-terminus of Cse4 (fused to MBP for expression). This new finding indicates that interactions of OA with the 601 DNA or with histones other than Cse4 made no significant contribution to the strength of the OA-NCP interface in our experiments (see Figure 5—figure supplement 1 and subsection “Both OA 271 and Mif2 form load-bearing attachments to centromeric nucleosomes”).

2) The authors may also want to re-think how they discuss their results in the background of the published yeast genetics data. While simultaneously adding Mif2 and OA to the Cse4 NCP gives increased strength of the attachment, the relation of this finding to native kinetochores is not straightforward: ΔN-Mif2 (defective in MIND binding) is viable, suggesting that the Mif2 based link of "force transmission" is not essential in yeast. Maybe the concept of different "paths of force transmission" through the kinetochore is not helpful at this point. Maybe distinct "load-bearing molecular links" within the kinetochore would be a better term. If the authors decide to revise this, they also need to change the title of the paper.

We sincerely thank the reviewers for bringing this to our attention. Although we knew this work well, we had not adequately incorporated it into our manuscript. In the revised version, we have deleted language suggesting that the Mif2-based path of force transmission is essential, when what we meant to say is that Mif2 is an essential protein. We have also explicitly cited the yeast genetics work (in the Discussion section of the revised manuscript), making clear that the N-terminus of Mif2 – and thus its ability to bind MIND – is dispensable for viability in budding yeast.

We stand by the concept of paths of force transmission, as it is a succinct descriptor for distinct chains of protein-protein interfaces between centromere and microtubules, all of which must be load-bearing. Because our work demonstrates that OA and Mif2 are independently capable of transmitting force from MIND to the centromeric nucleosome, we argue that this language is appropriate.

3) The discussion of what exactly the authors' assay measures should be improved. It was brought up whether it is indeed "mechanical strength" of a non-covalent bond that is measured in this assay. This would only be the case if the half-life of the respective bond was significantly longer than the time scale of the experiment (otherwise the assay is measuring the probability of the dissociation of the assembled complex in equilibrium with soluble components, determined by the thermodynamic dissociation rate constant). Is this condition met in the assay? The concentration of soluble subcomplexes is kept at 10 nM throughout. Isn't this an important experimental variable? Have the authors tried to vary the concentration? What is the effect in the binding and rupture assays?

We respectfully disagree with the assertion that mechanical strength of a non-covalent interaction can only be defined for bonds whose half-life is long compared to the measurement timescale. Bond strength is determined by both kinetic and thermodynamic properties and varies depending on the loading rate – even for simple cases involving just a single intermolecular bond. (See for example the studies of Evans and Ritchie, 1997, and Merkel et al., 1999, and the conceptual review by Bustamante et al., 2004.) Arrays of multiple binding elements, as found at kinetochores in vivo and in our in vitro reconstitutions, are more complicated of course. Nevertheless, their strength can also be defined and reflects both thermodynamic and kinetic properties. Importantly, there is no requirement that loading must be applied much more quickly than the typical kinetic rates for unbinding and re-binding of the individual elements. (See theoretical treatments of bond ‘clusters,’ Seifert, 2002, and Erdmann and Schwarz, 2004, for example.)

Testing concentration dependence will be an interesting challenge for the future. For the current study, we have chosen to keep the concentrations fixed in order to focus on how the strengths depend on which subcomplexes are included or omitted. Please note that for this study we have collected a total of 1,050 individual rupture events, measured across 19 different combinations of subcomplexes, which represents over a hundred hours of measurement time. Exploring the concentration dependencies for even one of these combinations will require a substantial amount of additional work, which we feel is beyond the scope of the present work. Our primary conclusion, that two distinct molecular chains connecting centromeric nucleosomes to microtubule tips can self-assemble and bear significant tensile loads, is well supported by the data collected at fixed concentrations. In the revised manuscript, we have clarified what our rupture force assay measures at the point in Results section at which it is introduced.

4) The authors should double-check their citations to make sure they are appropriate. Some problems that the reviewers identified were:- Subsection “OA forms microtubule attachments through MIND and Ndc80c”: The sentence deals with Ame1-MIND binding, but the cited Screpanti paper is about human CENP-C – Mis12 complex interaction.

We thank the reviewers for identifying this error. It has been corrected.

- Subsection “Mif2 forms microtubule attachments through MIND and Ndc80c”: The authors want to argue that parts of Mis2 other than the very N-terminus may contribute to MIND binding and cite Cheeseman et al., 2002. This is not covered in this publication, and the reviewers are not aware of other evidence.

We apologize for this error. We now cite Screpanti et al., 2011, who demonstrated that the region of conservation at the N-terminus of CENP-C/Mif2 extends beyond the first 40 amino acids. (See subsection “Mif2 forms microtubule attachments through MIND and Ndc80c”.)

- Discussion section: The sentence is about the different binding mode of OA and Mif2 to the centromeric Cse4-containing nucleosome, but the Straight lab papers that are cited deal with binding of vertebrate CENP-N and CENP-C to CENP-A containing nucleosomes, which does not directly support the argument.

Thank you for pointing out this potential confusion. We have clarified in the text that *homologs* of Mif2 have been shown to bind the C-terminus of homologs of Cse4, in contrast to OA, which has been shown to bind the N-terminus of Cse4.

- Subsection “Mif2 forms microtubule attachments through MIND and Ndc80c”: The binding interface is with Mtw1-Nnf1 (MN, head I), not with Dsn1-Nsl1. Please correct.

We thank the reviewers for pointing out this error and have corrected it.

- Discussion section: Yan et al., 2019, could be cited when discussing the weak interactions when using Cse4 NCPs assembled on 601 DNA and the potential role of centromeric DNA.

We thank the reviewers for this suggestion. Because the nucleosome structures that appear in Yan et al. were made with 601 DNA, we decided that this paper was not directly relevant to the question of whether or not centromeric DNA would strengthen interactions between the inner kinetochore and nucleosome core particles.

- Fischboeck et al. is still cited as Fischboeck et al., 2018 (bioRxiv), but should be Fischboeck et al., 2019 (eLife).

We apologize for the error and have corrected it.